

# Examining the role of red background in magnocellular contribution to face perception

Bhuvanesh Awasthi[1,2], Mark A. Williams[2] and Jason Friedman[3,4]

[1] Centre for Cognition and Decision Making, National Research University Higher School of Economics, Moscow, Russia
[2] ARC Centre of Excellence in Cognition and its Disorders, Department of Cognitive Science, Macquarie University, Sydney, Australia
[3] Department of Physical Therapy, Sackler Faculty of Medicine, Tel Aviv University, Tel Aviv, Israel
[4] Sagol School of Neuroscience, Tel Aviv University, Tel Aviv, Israel

## ABSTRACT

This study examines the role of the magnocellular system in the early stages of face perception, in particular sex categorization. Utilizing the specific property of magnocellular suppression in red light, we investigated visually guided reaching to low and high spatial frequency hybrid faces against red and grey backgrounds. The arm movement curvature measure shows that reduced response of the magnocellular pathway interferes with the low spatial frequency component of face perception. This finding provides behavioral evidence for magnocellular contribution to non-emotional aspect of face perception.

## INTRODUCTION

Perception of biologically relevant information from the environment is an essential feature of the primate sensory system. Humans are efficient and quick in detecting briefly viewed biological visual stimuli, faces in particular. In contrast to non-face objects, face perception is known to be carried out in a more configural manner, i.e, the information from the face is processed as an undifferentiated whole, at a single glance, beyond just the sum of the parts or the interrelations of the parts (*Young, Hellawell & Hay, 1987*; *Tanaka & Farah, 1993*; *Maurer, Le Grand & Mondloch, 2002*).

Different aspects of facial information processing have been associated with distinct ranges of spatial frequencies. For instance, configural processing may stem from low level sensory and perception mechanisms, particularly through Low Spatial Frequency (LSF) information, whereas High Spatial Frequency (HSF) conveys more local featural information about the stimulus (*Shulman et al., 1986*; *Hughes, Nozawa & Kitterle, 1996*; *Han, Yund & Woods, 2001*). The coarse-scale blurred features of LSF image may capture the diagnostic information needed for configural processing while fine-tuned details through HSF facilitates information regarding finer details of the stimuli.

Research on spatial frequency and visual processing has also established that LSF information is carried mainly via magnocellular channels that are structured for faster

Corresponding author
Bhuvanesh Awasthi,
bhuvanesh.awasthi@gmail.com

transduction of visual signals to the subcortical and cortical regions (*Livingstone & Hubel, 1988*; *Merigan & Maunsell, 1993*; *Bar, 2003*). Fine-grained High Spatial Frequency (HSF) information, on the other hand, has a comparatively slower transmission via the parvocellular channels (*Livingstone & Hubel, 1988*; *Bullier, 2001*). Magnocellular neurons project predominantly to the dorsal visual stream for information regarding motion and spatial location, while the recognition of object identity is processed along the ventral cortical pathway, mainly subserved by parvocellular channels (*Felleman & Van Essen, 1991*; *Orban, Van Essen & Vanduffel, 2004*). Finer details regarding the object's color, shape, inner feature and textures are carried chiefly through the slower parvocellular system, while the magnocellular system carries achromatic and global shape information (*Kaplan & Shapley, 1986*; *Macé, Thorpe & Fabre-Thorpe, 2005*). Parvo-outputs are thus predominant at higher spatial frequencies while Magno-outputs dominate at low spatial frequencies (*Tobimatsu, Tomoda & Kato, 1995*).

The magnocellular system facilitates rapid detection of object identity and location despite poor spatial resolution. Research findings illustrate that top-down modulations (predominantly through the LSF channels) bias the output of the bottom-up sensory input thereby facilitating rapid global processing (*Kveraga, Boshyan & Bar, 2007*; *Kveraga, Ghuman & Bar, 2007*). The global predictions obtained from the blurred LSF image are then progressively integrated with the HSF information in subsequent processes along the inferior temporal regions of the cortex. This reduces the computational demands on the system and improves efficient recognition.

Red light causes tonic suppression of the excitatory activity of (type IV) cells of the magnocellular pathway. This specific property of magnocellular suppression due to red color has been demonstrated in both animal and human studies throughout the visual stream including the retinal ganglion cells (*de Monasterio, 1978*), the lateral geniculate nucleus (*Dreher, Fukada & Rodieck, 1976*) and the striate cortex (*Livingstone & Hubel, 1984*). In recent times, several research reports suggested that red light suppressed the overall M pathway. Human behavioral studies have reported reduced performance on tasks biased toward predominant M pathway input in the presence of a red background (*Breitmeyer & Breier, 1994*; *Chapman, Hoag & Giaschi, 2004*; *Bedwell, Brown & Orem, 2008*; *Okubo & Nicholls, 2005*; *Bedwell et al., 2006*; *West et al., 2010*) on a variety of tasks such as reading performance (*Chase et al., 2003*), depth perception (*Brown & Koch, 2000*) and global motion (*Breitmeyer & Williams, 1990*).

Most research, so far, has been on how magnocellular pathway is involved in facial emotion processing (e.g. threat expressions, *West et al., 2010*) and emotion identification in schizophrenia (*Butler et al., 2009*; *Bedwell et al., 2013*). Emotion perception and face perception involve separate mechanisms (*Bruce & Young, 1986*; *Young & Bruce, 2011*). The role of M-pathway in non-emotional face processing remains to be investigated. *Laycock, Crewther & Crewther (2007)* have proposed a 'magnocellular advantage' model of visual perception wherein an initial rapid feedforward sweep through the dorsal stream activates parietal and frontal areas that further feeds back into the primary visual cortex. However, despite several

neuroimaging and neurophysiology reports, direct evidence for the role of the magnocellular system in the early stages of face perception remains to be established. Here, to examine the magnocellular contribution to face perception, we used a simple technique to selectively suppress the magnocellular system. We exploited the property of the magnocellular pathway's inhibitory response to red light (due to the long wavelength of red light).

By manipulating the background color, we designed the study to examine the contribution of the magnocellular pathway when processing LSF-HSF face hybrids. In the experimental task, there is greater emphasis on M-pathway input (i.e., identifying location, whether left or right) of the target rather than mere identification. Since the LSF interference is reliant on adequate magnocellular pathway functioning and the color red suppresses such functioning, we hypothesize that the curvature of trajectories with a neutral grey background is likely to be higher than those with the red background. Additionally, we predict a greater difference between the curvature between the fovea and periphery for the grey background compared to the red background, because red is likely to suppress the magnocellular input.

We use a relatively novel behavioral measure that provides rich information regarding perceptual decision making in real-time. Recent research reports (*Gold & Shadlen, 2001*; *Spivey & Dale, 2006*; *Freeman & Ambady, 2011*) have demonstrated that motor responses are generated in parallel with partial (incomplete) perceptual information that is continuously updated by perceptual-cognitive processing over time. Continuous tracking of hand movements during the reaching task can reveal otherwise hidden, underlying parallel processes in real-time (*Song & Nakayama, 2009*). Tracking of hand movements are reported to provide unusually high-fidelity, real-time access to fine-grained traces of the perceptual phenomena that are otherwise not captured by discrete traditional measures (*Resulaj et al., 2009*). Using a discrete measure (such as a button press reaction time response), we cannot disentangle with certainty and precision the early versus late processing of LSF and HSF in responding to the hybrids. Analyzing the shapes of the trajectories and separating the early versus late components of reaching movements allow early access to the state of the decision making process while the subjects reach to the targets.

There is a growing body of research that models how manual action exposes the real-time unfolding of underlying cognitive processing (*Song & Nakayama, 2009*; *Resulaj et al., 2009*). In our study, the reaching trajectories can potentially tease apart magno versus parvo inputs because in addition to the top-down recognition (of pointing to the target sex), the subjects are required to identify the location of target (whether left or right). In doing so, the trajectory they take (as the information in hybrids progressively unfolds) provides a window into the magnocellular afferents that project onto the dorsal stream structures important for reaching. Amongst a wide variety of measures that are used to examine the trajectories of movement (*Awasthi, 2012*), we use path offset/ curvature area in this study as this measure is well suited to provide valuable information based on task requirements and study design.

## METHOD

### Ethics statement

The ethical aspects of this study were approved by the Human Research Ethics Committee (HREC) of Macquarie University (approval number HE23NOV2007-R05540).

### Stimuli, design and procedure

Similar to the previous experiments (described in *Awasthi, Friedman & Williams (2011a)* and *Awasthi, Friedman & Williams (2011b)*), male and female face images were Fourier transformed and multiplied by low-pass and high-pass Gaussian filters to preserve low (<8 cpf) and high SF (>25 cpf) information in each image. The images were equated for luminance and contrast energy using Shine toolbox before filtering and were then superimposed to make LSF-HSF hybrid images (see Fig. 1). Both the grey (RGB values of 75, 75, 75) and red background (RGB values of 255, 0, 0) had equal luminance. Presentation software (Neurobehavioral Systems, Albany, CA, USA) was used to present the stimuli. The stimuli had a mean height and width of 5.7° visual angle at foveal presentation and were presented 21.7° from fixation for peripheral presentation.

Four combinations of hybrid images were used in the experiment. A four-factor within-subjects design was used, the factors being Eccentricity (periphery, fovea), Target Location (left, right), Target Congruity (congruent, incongruent) and Distractor Conflict (present, absent). All factors were fully crossed, yielding sixteen experimental conditions. Congruity was defined as the sex of the HSF face being the same as that of the LSF face in a hybrid. Thus, MM and FF were congruent whereas FM and MF were incongruent conditions. For instance, in the hybrid image **FF**, the first letter (**F**) of the hybrid indicates the sex of the LSF face (**Fem**ale) and the second letter (**F**) indicates the sex of the HSF face (**F**). For hybrid **FM**, the LSF face is **F**emale and the HSF face is **M**ale (Fig. 1a).

For each subject, the target sex was assigned at the beginning of the experiment for the entire session (e.g., Female). The task involved pointing to the target (e.g., female face) on the touch screen. At viewing distance, the HSF face was salient (most visible) and was the target for all trials throughout the experiment. In congruent target trials, both the LSF and HSF components of the hybrid face were the same sex (e.g., both Female; FF) while for the incongruent target trials, the LSF component was of the opposite sex (e.g., Male; MF). To maximize the LSF interference effect, we also manipulated whether the face on the other side of the target location held a target-matched LSF distractor (e.g., an LSF female) or not (e.g., no LSF female). There was no HSF distractor. Two hybrid faces were presented at the left and right side of the fixation cross (for foveal presentation) and at far periphery (for peripheral presentation) (Fig. 1b). Subjects were instructed to begin the task with eyes on the fixation. After the initial liftoff, they could move their eyes freely.

### Subjects

Right-handed subjects were recruited from the Macquarie University community who volunteered their participation. All subjects reported normal or corrected-to-normal vision and gave written, informed consent before participation. Fifteen subjects (seven females, age range: 18–30, mean age: 25.2 years, SD = 4.3) participated in the task.

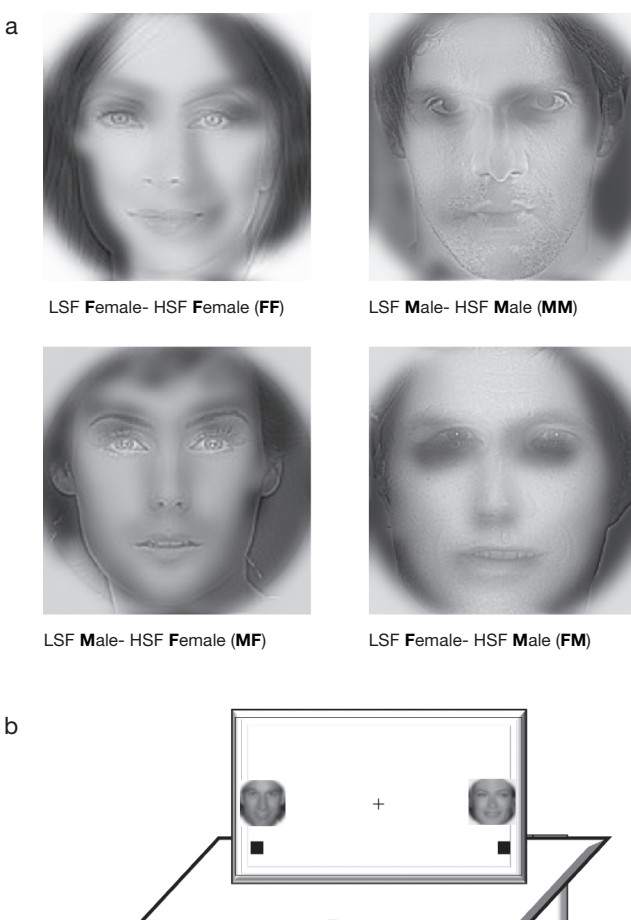

**Figure 1** (A) Stimuli: LSF-HSF hybrid images used in the experiments. To see the LSF content, squint, blink, or step back from the figure (B) Experimental setup showing the touchscreen where two hybrid images were presented peripherally and centrally in alternate blocks. Subjects start each trial at the grey button on the table to reach out and touch the respective black target box on the screen.

## Procedure

Subjects sat in a quiet, dark room at a table with a LCD touch screen (70 × 39 cm, 1360 × 768 pixels, 60 Hz) positioned approximately 70 cm in front of them. Two small infrared light emitting diode (LED) markers were attached to the right index fingertip of the subject. The starting position (a button) was aligned with the body midline, approximately 20 cm in front of the subjects. Each trial began with subjects placing their right index finger on the centrally located starting button in front of the touchscreen. Hand movements were tracked with an Optotrak Certus Motion Capture System (Northern Digital Inc., Waterloo, Ontario, CAN) at a 200 Hz sampling rate. The tracking system was calibrated at the beginning of each experiment.

In both the grey background and the red background conditions, two hybrids were presented *peripherally* and *centrally* in alternate blocks. The peripheral and central

presentation of hybrids is equivalent in terms of stimuli and the procedure, except that in the central condition, two hybrids were present on either side of fixation. Subjects performed the red and grey sessions on consecutive days in a counterbalanced manner. They were assigned Male or Female as the target sex (also counter-balanced across subjects). They were instructed to maintain fixation on a cross at the center of the screen (that appeared for 1000 ms) before reaching out and pointing to the target. Subjects had to begin their reaching response within 350 ms of target onset but their final responses (either left or right) were not speeded, with sufficient (cutoff 1.5 seconds) time for the finger to change direction or correct its course. This enabled an index of the early stages of perception since subjects' initial movements can provide a measure of information accumulation that is otherwise lost in later final decision indicators such as response times. The specific choice of 350 ms was based on previous studies (*Awasthi, Friedman & Williams, 2011a*; *Awasthi, Friedman & Williams, 2011b*; *Awasthi, Friedman & Williams, 2012*), and was selected to be significantly less than typical reaction time for this sort of task (*Goffaux & Rossion, 2006*).

Trials were aborted when started too early (before the target onset) or too late (after 350 ms). For all responses, feedback was provided onscreen. All conditions were presented in each block, consisting of 48 trials (3 trials of each of the 16 types) in a pseudo-randomized order. After a training block, 16 additional blocks were carried out breaks in between. Only the correct response trials were used for further data analysis. The subjects had a mean accuracy rate of 92.67 % (SD = 3.75) for the grey background and 94.14% (SD = 3.63) for the red background. The accuracy was not significantly different across the two background colors (F (1, 14) = 1.26, p = 0.28, partial $\eta^2$ = 0.083).

We used cubic splines for data smoothing and interpolation when markers were occluded (for less than 10% of the trajectory in a trial). Movement data was analyzed using Matlab (The Mathworks, Inc). We calculated the maximum deviation from a straight-line path from start to end of the movements. We then defined maximum curvature as the ratio of this deviation to the length of the straight-line path (*Atkeson & Hollerbach, 1985*; *Smit & Van Gisbergen, 1990*). The average maximum curvature was computed for all subjects and used as the dependent variable.

## Curvature results

The curvature in the trajectory is taken as a measure of uncertainty in the decision making process. The mean trajectories (pooled for all subjects) are shown in Fig. 2. The differences in the trajectories were initially quantified by comparing the maximum curvature. When subjects select a target and do not change their mind (i.e., when the LSF and HSF components unambiguously correspond to one target), they reach directly to the target, and the curvature is low. When there are competing sources of information (i.e., the LSF and HSF are incongruent, or there is an LSF distractor on the other side), this uncertainty in the decision making process is reflected through greater curvature. Repeated measures ANOVA with background color (grey, red), eccentricity (periphery, fovea), target conditions (congruent target-no distractor, incongruent target–no distractor, congruent

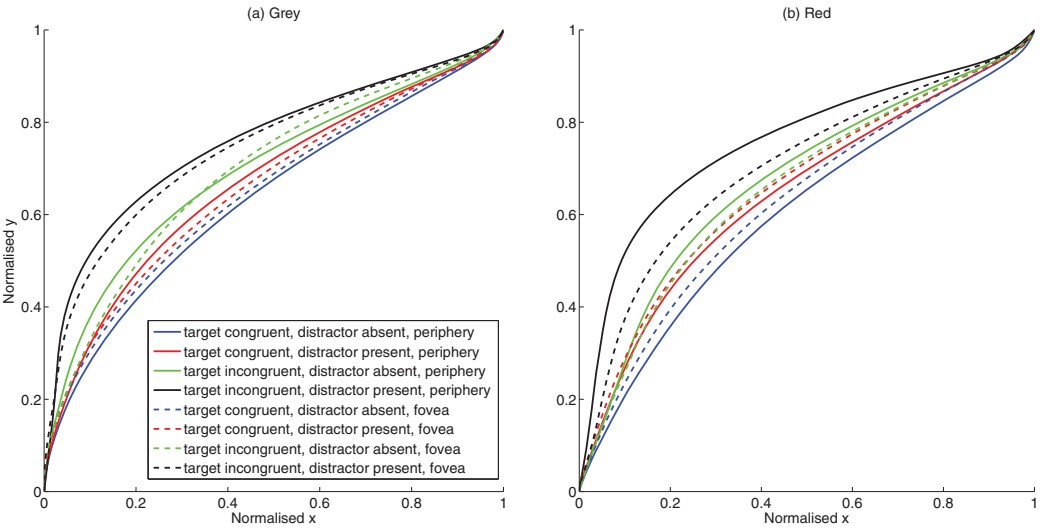

**Figure 2 Trajectory plots.** (A) Mean trajectories for grey background (B) Mean trajectories for red background.

target–distractor present, incongruent target–distractor present) was carried out. Tests of violations of sphericity were performed on the data.

Eccentricity and target condition significantly influenced the curvature as shown by the significant main effect of eccentricity $F(1, 14) = 13.60$, $p = 0.002$, partial $\eta^2 = 0.493$) and target condition $F(3, 12) = 110.67$, $p < 0.001$, partial $\eta^2 = 0.888$). Curvature was significantly larger (mean difference = 0.021, CI.$_{95}$: 0.009, 0.034) for presentations at the periphery (mean = 0.265) than for fovea (mean = 0.244). Conflict due to congruity and distractor conditions also resulted in larger curvature [(Congruent target No distractor mean = 0.210; Incongruent target No distractor mean = 0.235; mean difference = 0.026, CI.$_{95}$: −0.032, −0.019) (Congruent target distractor present mean = 0.272; Incongruent target distractor present mean = 0.301; mean difference = 0.029, CI.$_{95}$: −0.042, −0.016)]. The main effect of background color also approached significance $F(1, 14) = 3.27$, $p = 0.092$, partial $\eta^2 = 0.190$) with pairwise comparisons showing larger curvature for grey (mean = 0.272) than red (mean = 0.237) background (mean difference = 0.035, CI.$_{95}$: 0.007, 0.077).

The effect of target conditions on curvature is significantly larger for grey than red background as shown by a significant interaction between background color and target conditions $F(1, 14) = 11.88$, $p < 0.001$, partial $\eta^2 = 0.459$). Post-hoc (Tukey HSD) analysis showed that curvature differences for grey background presentations were significantly greater for congruent targets (mean difference = 0.043, CI.$_{95}$: 0.005, 0.080) in grey (M = .231) than for the red (M = 0.188) background (p = 0.02) (see Fig. 3a).

Curvature differed across eccentricity when conflict (incongruity or distractor) was present as shown by a significant interaction between eccentricity and target conditions $F(3, 12) = 19.64$, $p < 0.001$, partial $\eta^2 = 0.584$). Post-hoc (Tukey HSD) analysis showed that difference in trajectories across periphery and fovea were significantly different for conditions with conflict (incongruity and distractor presence) conditions. For instance,

(a) Background color X Target conditions

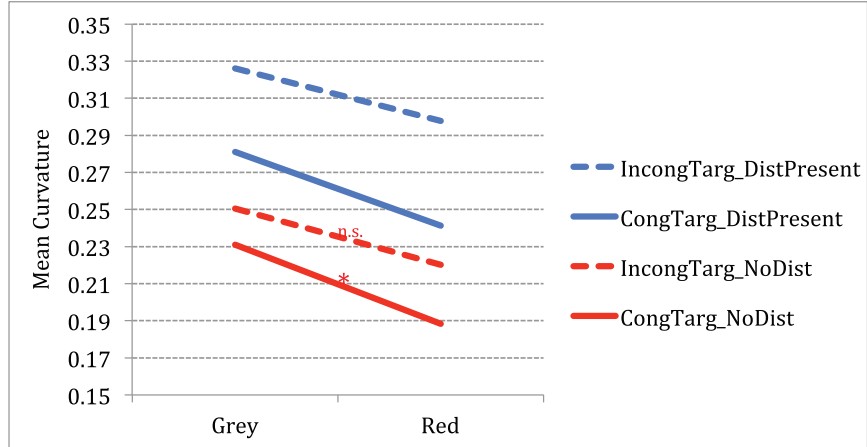

(b) Eccentricity X Target conditions

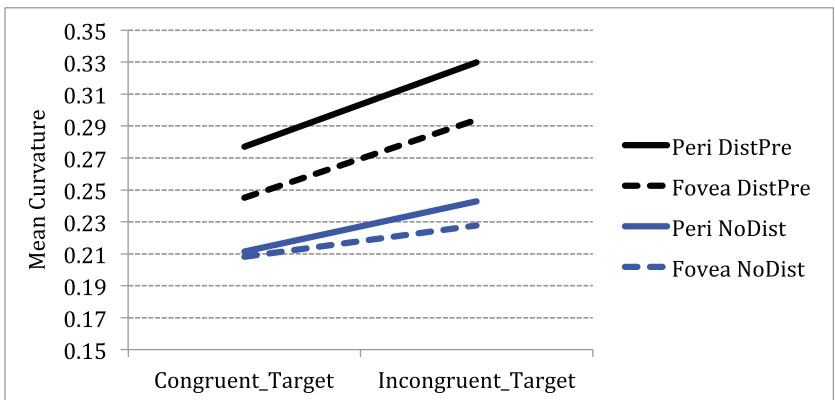

(c) Background Color X Eccentricity X Target conditions

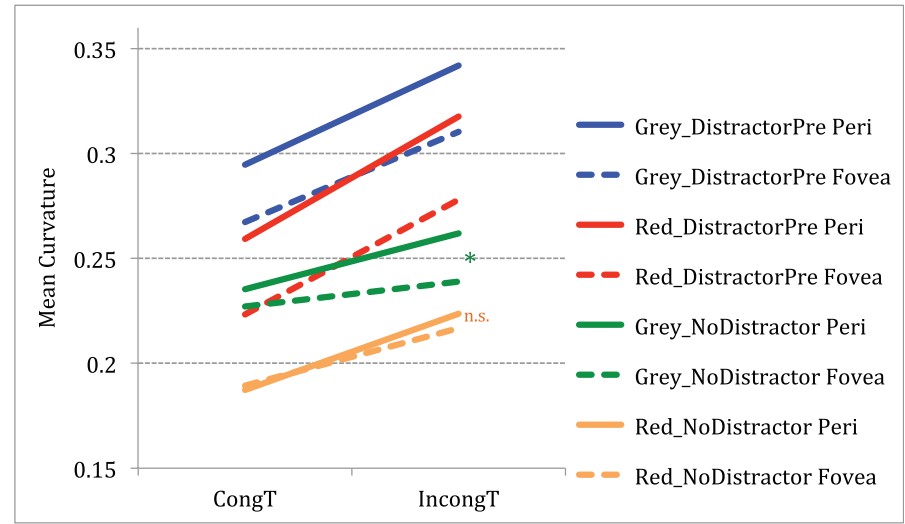

**Figure 3 Graphs showing significant interactions between conditions.** (A) Significant 2-way interaction between Background color × Target conditions (B) Significant 2-way interaction between Eccentricity × Target conditions (C) Significant 3-way interaction between Background color × Eccentricity × Target conditions.

curvature differed significantly for periphery versus fovea for incongruent- no distractor conditions (mean difference = 0.015, $CI_{.95}$: 0.0009, 0.0290) at $t(14) = 2.28$, $p = 0.03$; for congruent target distractor present conditions (periphery versus fovea mean difference = 0.031, $CI_{.95}$: 0.0165, 0.0467) at $t(14) = 4.50$, $p < 0.001$ as well as for incongruent target distractor present conditions (periphery versus fovea mean difference = 0.035, $CI_{.95}$: 0.0161, 0.0551) at $t(14) = 3.92$, $p = 0.002$.

Further, results show that curvature differences across eccentricity for congruent targets were significantly larger in grey but not in red background, as shown by a three-way interaction between background color × eccentricity × target conditions $F(3, 12) = 9.002$, $p < 0.001$, partial $\eta^2 = 0.391$). Post-hoc (Tukey HSD) analysis showed that the curvature for congruent target no distractor conditions across fovea and periphery were significantly larger for the grey background than for red background (grey versus red at periphery (mean difference = 0.048, $CI_{.95}$: 0.003, 0.092)) at $t(14) = 2.307$, $p = 0.037$; grey versus red at fovea (mean difference = 0.038, $CI_{.95}$: 0.003, 0.072)) at $t(14) = 2.353$, $p = 0.034$, effectively showing suppression in red. In addition, for the incongruent target no distractor conditions, there was significant difference between periphery and fovea for grey (mean difference = 0.023, $CI_{.95}$: 0.006, 0.039) at $t(14) = 2.96$, $p = 0.01$, but not for red background (mean difference = 0.007, $CI_{.95}$: 0.012, 0.026) at $t(14) = 0.783$, $p = 0.447$ (see green and orange lines in Fig. 3c).

To quantify the sensitivity of the statistical tests described above, we also carried out power analyses as demonstrated in detail by *Campbell & Thompson (2012)*. For this, we used the software MorePower 6.0.1 (https://wiki.usask.ca/pages/viewpageattachments.action?pageId=420413544). Calculations were performed using the F values and MSE (Mean square error) from SPSS with repeated measures 2 × 2 × 4 design. We find that all of our reported main effects (eccentricity, target conditions) as well as interactions (background × target conditions; eccentricity × target conditions and background × eccentricity × target conditions) showed robust effects (see Table 1). As we can see, the sample size is sufficient to detect a significant effect at the specified alpha level (alpha = .05) for all the contrasts of interest described above.

## DISCUSSION

In this study, we examined the role of LSF and HSF in a face categorization task at fovea and periphery. To infer the role of the magnocellular visual pathway, a novel technique was used wherein the reaching performance against a red background (believed to suppress the magnocellular pathway) was compared to that against a neutral grey background. Results from curvature measure demonstrate the differential role of grey and red background in sex categorization task, likely indicating the magnocellular suppression in red. Specifically, larger differences were observed between conditions when the background was grey compared to the red background, corresponding to the first prediction. Additionally, for some cases, the incongruence effect was greater at the periphery than at fovea, only for the grey and not the red background, corresponding to the second prediction. Early information in LSF faces is carried through the magnocellular pathway and this information may be crucial for quick start times required in the fast categorization task used here.

**Table 1 Results of the power analysis for all conditions.** Results of the power analysis using More-Power 6.0.1 wherein for the repeated measures 2 × 2 × 4 design, calculations were performed using the F values and MSE (Mean square error), showing the Power for all conditions.

| Parameter | Levels | F | MSE (mean square error) | Power |
|---|---|---|---|---|
| Background color | 2 | 3.274 | 0.075 | 0.392 |
| Eccentricity | 2 | 13.608 | 0.027 | 0.929 |
| Target conditions | 4 | 110.67 | 0.097 | 1 |
| Background × Eccentricity | 2 × 2 | 0.04 | 0.0000876 | 0.05 |
| Background × Target conditions | 2 × 4 | 11.881 | 0.008 | 0.999 |
| Eccentricity × Target conditions | 2 × 4 | 19.642 | 0.009 | 0.999 |
| Background × Eccentricity × Target conditions | 2 × 2 × 4 | 9.002 | 0.005 | 0.992 |

Previous research has implicated the possible role of the magnocellular pathway in rapid detection and categorization of achromatic stimuli (*Delorme, Richard & Fabre-Thorpe, 1999*). For example, *Michimata, Okubo & Mugishima (1999)* showed that red background impairs the perception of a global visual pattern believed to rely on LSF processing. Other studies have shown reduced flicker sensitivity when presented on a red background (*Stromeyer, Cole & Kronauer, 1987*). *West et al. (2010)* also demonstrated that the perception of fearful faces is suppressed under red diffused light, lending support to the idea of global precedence through the magnocellular pathway.

In our previous studies using the same experimental setup (*Awasthi, Friedman & Williams, 2011a*; *Awasthi, Friedman & Williams, 2011b*), we demonstrated significantly larger peripheral interference by LSF information while reaching for HSF targets. In concordance with our previous findings and the research showing higher density distribution of magnocellular channels at periphery, the LSF interference in the current study reflects the differential engagement of the magnocellular pathway in grey versus red background. In yet another study using red glasses for experimental manipulation, *Williams, Grierson & Carnahan (2011)* demonstrated that diffused red light suppresses magnocellular activity thereby significantly disrupting visual threat processing.

*Skottun (2004)* has argued that red light also possibly attenuates the parvocellular red–green color-opponent cells. In contrast, studies have demonstrated that color-contrast detection by red-green opponent cones is confined to foveal vision and is known to fall steeply across the periphery (*Mullen & Kingdom, 2002*; *Mullen, Sakurai & Chu, 2005*), specifically >20 degrees of eccentricity. Our experimental results do not seem to show parvocellular disruption as the accuracy remained high for HSF targets with red background, especially at peripheral locations >21 degrees of visual angle. Peripheral inputs are served predominantly through magnocellular pathway, mostly via LSF information (*Dacey & Petersen, 1992*; *Silveira & Perry, 1991*). However, it is probable that the red background interacts with the parvocellular pathway as well, mechanisms of which remain to be investigated. It is also not unlikely that there is some role of the koniocellular (K) pathway that sends projections to the primary visual cortex (*Cassagrande, 1994*) details for which are as yet unknown. A potential limitation here is the relatively small sample size

(15 subjects) compared to the number of conditions analyzed ($2 \times 2 \times 4 = 16$ conditions). However, as our conclusions are based on the highly significant findings, it appears that the power is sufficient. The results of the power analysis show that the main effects and interactions are robust. In future experiments, a larger sample size would help to determine whether there are more significant differences other than those reported here.

Across all the red diffuse light literature, there is no standard of luminance; instead there is a standard of keeping equiluminance across all conditions. It may be the case that researchers who use high luminance may see a greater effect of inhibition of the M-pathway than those who use lower luminance. However, researchers have been able to find the effects of red diffuse light with both high luminance (*Chapman, Hoag & Giaschi, 2004*), and low luminance (*Mullen & Kingdom, 2002*). Research by Breitmeyer has shown that dark adaptation was useful to observe the inhibitory effects of red diffuse light (*Breitmeyer & Breier, 1994*; *Breitmeyer & Williams, 1990*) by providing the required bias towards the M-pathway.

Deficits in global identification of faces (as seen in prosopagnosia) and other developmental conditions might have a magnocellular contribution (*Grinter, Maybery & Badcock, 2010*). It is likely that magnocellular dysfunction, if present in face recognition deficits, concerns more integrated processes at levels where information from parvocellular and magnocellular channels interact. In line with (*Laycock, Crewther & Crewther, 2007*) magnocellular deficit proposal, we recently reported delayed integration of Low and High Spatial Frequency information in developmental prosopagnosia (*Awasthi, Friedman & Williams, 2012*). This integration failure could be due to an attention-perception mechanism that favors parvocellular over magnocellular channels reflecting a tradeoff between segregation and integration of information (*Yeshurun, 2004*) that ultimately reflects in impaired face processing. The deficient responsiveness of the magnocellular channels has been reported in schizophrenia (*Martinez et al., 2008*) as well as in other developmental conditions (*Laycock, Crewther & Crewther, 2007*), subsequently influencing higher level perceptual processing.

Findings from our study provide strong support for the role of the magnocellular pathway in face perception. The contextual associations between stimuli in the environment activate predictive early guesses, available through partial, blurred, yet rapidly available information about the stimulus identity. It is noteworthy though that non-face object may also utilize this magnocellular advantage. However, much experimental data on the topic remains to be generated to generalize the advantage to certain or all object classes. *Fenske et al. (2006)* demonstrate the top-down facilitation that is triggered by both the early information about an object, as well as by contextual associations between an object and other objects in the vicinity within which it is typically recognized. As the magnocellular pathway is predominantly attuned to Low Spatial Frequency (LSF) information, Bar's proposed model predicts that the early, quick and dirty LSF information processing is projected to the orbito-frontal cortex, which in turn selects potential matches based on the global, LSF-based properties of the bottom-up input. Predictions to identity fine-tuned features of the stimulus global properties (obtained initially from the blurred LSF image) are then projected to the high spatial

frequency dominated object recognition areas along the inferior temporal regions of the cortex.

Taken together, it may be more appropriate to describe the predictions generated by the low and high SF as phases evolving over time, rather than as separate bouts of visual information (although using separate pathways) each one adding further to the perceptual processing. The magnocellular system helps in deciphering the overall three-dimensional organization, position and movement of visual objects in the environment. Selective engagement of the magnocellular system provides a framework for expedient prioritization of biologically salient stimuli that compete for attention and action.

### Funding
The authors received no funding for this work. The funders had no role in study design, data collection and analysis, decision to publish, or preparation of the manuscript.

### Competing Interests
The authors declare that they have no competing interests.

### Author Contributions
- Bhuvanesh Awasthi conceived and designed the experiments, performed the experiments, analyzed the data, contributed reagents/materials/analysis tools, wrote the paper, prepared figures and/or tables, reviewed drafts of the paper.
- Mark A. Williams contributed reagents/materials/analysis tools, wrote the paper, reviewed drafts of the paper.
- Jason Friedman conceived and designed the experiments, analyzed the data, contributed reagents/materials/analysis tools, wrote the paper, prepared figures and/or tables, reviewed drafts of the paper.

### Human Ethics
The following information was supplied relating to ethical approvals (i.e., approving body and any reference numbers):

Human Research Ethics Committee (HREC) of Macquarie University.

Approval number HE23NOV2007-R05540.

### Data Deposition
Figshare: https://dx.doi.org/10.6084/m9.figshare.1083788.v2.

### Supplemental Information
Supplemental information for this article can be found online at http://dx.doi.org/10.7717/peerj.1617#supplemental-information.

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
