# Peer review of "Examining the role of red background in magnocellular contribution to face perception"

_PeerJ, doi:10.7717/peerj.1617_

## Round 0.1 · original submission · Major Revisions

The reviewers have major concerns about the effect size and statistical evidence concerning significance. This stems from the use of inappropriate statistical methods. Please redo all the statistics that take into account the power of your design, so that the source of variance can be identified, and significance can be truly assessed. Note that this is a methodological issue that will preclude publication in PeerJ if not addressed correctly.

·

Basic reporting

Abstract
It is an overstatement to claim that this is the first “definitive” behavioral evidence. Red light also has effects on the P pathway and cannot be used to definitely manipulate only the M pathway. It just “suggests” M pathway involvement, but not necessarily more so than other existing behavioral studies on this topic. This statement should be removed and replaced with a more accurate statement that it adds to the literature suggesting the M pathway involvement in facial perception.
Lines 110-111 – these later studies were not single cell recording studies and only suggested that red light suppressed the overall M pathway (they are not directly comparable to Wiesel and Hubel as sentence suggests).
Line 117 – more accurate to say tasks that are biased toward predominant M pathway input (P pathway is still involved)
Lines 121-123. The Bedwell et al (2006) study did not find overall suppression in V5 in controls from red light. Instead it found that the proportion of activity in the right hemisphere V5 as compared to the sum of both V5s was reduced in controls. This is an important distinction as it only suggests a shift in laterality.
In the hypotheses, authors need to clarify what the expected effect of LSF interference on trajectories is (with a neutral background)

Experimental design

The authors use a 5 factor ANOVA for the primary analysis with a total of 15 participants. This very small number of participants does not even approach adequate statistical power for a 5 factor ANOVA. All results are therefore suspect and likely invalid.

Other more minor points:
Section 2.3 – Authors should state the range of the age.
Line 234 – Authors need to clarify why 350 ms was chosen as the cutpoint for maximum start of reaching response. What is that decision based on?
Line 238 – Were the 16 experimental conditions presented in a blocked manner across these 16 blocks? If so, was the order of these 16 conditions randomized by participant? How long were the breaks? If conditions were fully randomized across these blocks, why were 16 blocks created? Why were only correct trials analyzed (what is the rationale for that decision)?
Lines 240-241 – Authors should clarify if the accuracy between colors was significantly different – and report the resulting stat values, including effect size.

Validity of the findings

As noted above, I do not believe that the statistical analyses can be interpreted given the extremely low statistical power that results from a 5 factor ANOVA with only 15 participants.

Reviewer 2 ·

Basic reporting

This manuscript addresses whether or not magnocellular pathway contributes to LSF component of facial recognition. The hypothesis is very interesting. If true, it would provide important insight into the role of M pathway in object recognition.
However, the present data do not convincingly support the conclusion that M pathway actually contributes to facial recognition for the following reasons.
1. “Further results show that congruity differences at fovea for grey background (M= 0.27) were significantly larger than for the red background (M=0.20) but not for the periphery” (280). This result is against their hypothesis that “the effect on trajectories is likely to be less pronounced at peripheral locations with a red background compared to a grey one” (140). It actually suggests that the effect of red background is likely caused by the P pathway since the P pathway is more predominant at fovea and could be affected by the red background as well (310-315).
2. “Mean curvature due to target congruity was significantly larger in peripheral presentations (M= 0.22) than in the foveal presentations (M=0.30)”(276-277). This result can also be explained by a relatively larger contribution from LSF component of the P pathway at peripheral due to the decrease in the preferred SF toward peripheral.
To address the above concerns, the effects of red background and eccentricity on the P pathway need to be tested and/or discussed.

Minor comments:
1. “Mean curvature due to target congruity was significantly larger in peripheral presentations (M= 0.22) than in the foveal presentations (M=0.30)”(276-277). The statement is inconsistent with the number.
2. All statistics are presented in sentences. It would be helpful to present a few important ones in graphs.

Experimental design

No comments

Validity of the findings

Not convincing since the results could be explained by the contribution from the P pathway

Additional comments

No comment

---

## Round 0.2 · Major Revisions

The study has been seen by the same two reviewers who remain concerned about the scholarly presentation of the work, and one reviewer noted remaining methodological (statistical) flaws. Note that, although PeerJ policy does allow for publication of replications, it will not publish methodologically flawed studies. It seems to me that the study might be strengthened by running more subjects to improve the statistical power. Any revised MS will be returned to the same reviewers so please clearly address each and every concern in the point-by-point response.

·

Basic reporting

The Introduction claims that the role of the M pathway in face perception is still unclear. However, there is already a large body of literature that supports the role of the M pathway in face perception - one of which used the red light manipulation (West et al., 2010). Also, see studies such as Butler PD et al 2009 Schiz Bulletin - nonpsychiatric control data. A more thorough literature review on what is already known about the role of the M pathway in facial processing is needed, and then a clarification on how this design is going to add to that existing knowledge. It is not clear to me that there is a need for a study to replicate the existing knowledge that the M pathway contributes to the processing of any image - including faces. The nuances of "how" it contributes, beyond what is already known, is still needed, but the authors need to detail how this design is able to accomplish that.

Experimental design

The authors responded to my concern about very low statistical power by changing from a 5 factor model to a 3 factor model. This does not address my concern, as it appears that they just created more levels under one of the factors. Overall, this study has 16 conditions and 15 participants - which results in low statistical power that may explain some of the null findings (e.g., in high contrast condition).

Validity of the findings

The second reviewer makes good points about the P pathway also responding to red. The P pathway is very sensitive to color, including red. There are still some cones in the periphery although they are relatively fewer in number than fovea. Therefore, the authors cannot make such strong claims that it must be due to the M pathway (e.g., in current abstract and throughout). The behavioral results can only "suggest" such things - so language needs to be toned down (still) throughout, including abstract. In terms of validity, my point about the study being notably statistically underpowered lowers confidence in the validity of all of the findings.

Additional comments

The study would greatly benefit from running more participants to reach an acceptable level of statistical power for the number of experimental conditions.

Reviewer 2 ·

Basic reporting

The addition of Fig. 3 and more detailed discussion on the P pathway address some of previous concerns. However, the following results seem to be inconsistent with the main conclusion and need to be clarified.
1. If the difference in curvature between congruent and incongruent was caused by the M pathway, which is presumably suppressed by red background, the difference should be greater for gray background than for red one. However, both Figs 3a and 3c show the opposite.
2. For congruent stimuli, the LSF is supposed to facilitate the reach. If the LSF is carried by the M pathway, suppression of it by red light should remove the facilitation and increase the curvature. But once again, both Figs 3a and 3c show the opposite.
It would be helpful to list predictions of the main hypothesis and discuss which one is consistent or not with the results, and the implication in terms of supporting or refuting the hypothesis.
In the rebuttal, the authors state “According to the literature, peripheral input is not well served by parvocellular pathway. Peripheral inputs are served predominantly through magnocellular pathway, mostly via LSF information”. It is better to include this statement in the manuscript with proper citation.

Experimental design

No Comments

Validity of the findings

No Comments

---

## Round 0.3 · Major Revisions

The previous reviewers were unavailable to review so I have reviewed the work myself. My feeling is that you have responded well to the reviewers concerns but have not gone far enough to address the very serious statistical concerns of one reviewer. In your rebuttal you conclude (I'm paraphrasing here) that it's ok to stick with an N=15 since your claims relate to only those findings that are significant and (therefore to your logic) powered appropriately. This logic does not follow in that the concern is that some results may potentially be spurious. Further, some of the insignificant results could potentially have changed your conclusions had they been significant. So I propose that you address these issues by performing a power analysis. If the power is higher than 0.8, your conclusions are reasonable. If lower, than you must concede that your study may be underpowered. Even if that is the case, it does not preclude publication (without adding N) so long as you are very clear of your study's weaknesses and the implications of a more highly powered study that revealed new significant findings. Adding subjects would be even better, in such a case.

---

## Round 0.4 · accepted · Accept

Thank you for providing the new power analyses. The manuscript is now ready for publication.